# Role of miR-30a-3p Regulation of Oncogenic Targets in Pancreatic Ductal Adenocarcinoma Pathogenesis

**DOI:** 10.3390/ijms21186459

**Published:** 2020-09-04

**Authors:** Hiroki Shimomura, Reona Okada, Takako Tanaka, Yuto Hozaka, Masumi Wada, Shogo Moriya, Tetsuya Idichi, Yoshiaki Kita, Hiroshi Kurahara, Takao Ohtsuka, Naohiko Seki

**Affiliations:** 1Department of Digestive Surgery, Breast and Thyroid Surgery, Graduate School of Medical and Dental Sciences, Kagoshima University, Kagoshima 890-8520, Japan; jaga-rosso@hi.enjoy.ne.jp (H.S.); chotochu.taka@gmail.com (T.T.); k6958371@kadai.jp (Y.H.); masumi8823@yahoo.co.jp (M.W.); idichitetuya@gmail.com (T.I.); north-y@m.kufm.kagoshima-u.ac.jp (Y.K.); h-krhr@m3.kufm.kagoshima-u.ac.jp (H.K.); takao-o@kufm.kagoshima-u.ac.jp (T.O.); 2Department of Functional Genomics, Chiba University Graduate School of Medicine, Chiba 260-8670, Japan; reonaokada@chiba-u.jp; 3Department of Biochemistry and Genetics, Chiba University Graduate School of Medicine, Chiba 260-8670, Japan; moriya.shogo@chiba-u.jp

**Keywords:** pancreatic ductal adenocarcinoma, tumor suppressor, pathogenesis, microRNA, miR-30a-3p

## Abstract

Our recent studies have implicated some passenger strands of miRNAs in the molecular pathogenesis of human cancers. Analysis of the microRNA (miRNA) expression signature in pancreatic ductal adenocarcinoma (PDAC) has shown that levels of miR-30a-3p, the passenger strand derived from pre-mir-30a, are significantly downregulated in PDAC tissues. This study aimed to identify the oncogenes closely involved in PDAC molecular pathogenesis under the regulation of miR-30a-3p. Ectopic expression assays showed that miR-30a-3p expression inhibited the aggressiveness of the PDAC cells, suggesting that miR-30a-3p acts as a tumor-suppressive miRNA in PDAC cells. We further identified 102 putative targets of miR-30a-3p regulation in PDAC cells by combining in silico analysis with gene expression data. Of these, ten genes (*EPS8*, *HMGA2*, *ENDOD1*, *SLC39A10*, *TGM2*, *MGLL*, *SERPINE1*, *ITGA2*, *DTL,* and *UACA*) were independent prognostic factors in multivariate analysis of survival of patients with PDAC (*p* < 0.01). We also investigated the oncogenic function of the integrin *ITGA2* in PDAC cell lines. The integrin family comprises cell adhesion molecules expressed as heterodimeric, transmembrane proteins on the surface of various cells. Overexpression of ITGA2/ITGB1 (an ITGA2 binding partner) was detected in the PDAC clinical specimens. The knockdown of *ITGA2* expression attenuated the malignant phenotypes of the PDAC cells. Together, results from these microRNA-based approaches can accelerate our understanding of PDAC molecular pathogenesis.

## 1. Introduction

Ductal carcinoma of the pancreas (PDAC) is derived from exocrine cells of the pancreatic duct and accounts for 90% of pancreatic cancer cases [1,2,3]. Due to the aggressive phenotype of PDAC cells, which includes high invasiveness and drug resistance, the 5-year relative survival rate for pancreatic cancer is only 3–10%, which is one of the lowest among human cancers [1,4]. Since many patients with PDAC are asymptomatic, most cases have reached an advanced stage at the time of diagnosis.

Current standard treatments for PDAC still consist of surgical resection and cytotoxic chemotherapies, although fewer than 20% of PDAC patients are candidates for complete surgical resection [1,5]. Gemcitabine is an essential drug for the treatment of PDAC. FOLFIRINOX (folinic acid, 5-FU, irinotecan, and oxaliplatin), as well as gemcitabine and nab-paclitaxel regimens, are given as first-line treatments for advanced PDAC with distant metastases. However, these treatment regimens typically offer insufficient therapeutic effects [6]. Thus, new diagnostic markers and the development of new therapeutic strategies based on the latest genomic analyses are needed for patients with PDAC.

In the post-human genome era, it has become clear that non-coding RNAs are functional and control pivotal biological functions in a wide range of biological processes [7,8]. MicroRNAs (miRNAs) are endogenous non-coding RNAs that are single-stranded molecules having lengths between 19 and 23 nt that function to fine-tune RNA expression control via degradation or translational inhibition of target RNA transcripts [7,8,9]. A single miRNA can control a large number of RNA transcripts for both protein-coding genes and non-coding genes. Expressions of around 60% of cellular RNAs are controlled by miRNAs [10,11]. Therefore, the aberrant expression of miRNAs can disrupt RNA networks and trigger the transformation of normal cells to diseased cells.

In the last decade, multiple studies have shown that aberrant expressions of miRNAs are involved in the molecular pathogenesis of human cancers and that their dysregulated expressions can enhance cancer cell aggressiveness via several pathways, such as loss of cell cycle regulation, suppression of apoptosis, promotion of cell invasion and metastasis, and acquisition of drug resistance [12,13,14]. In miRNA-based cancer research, the starting point of many studies is the identification of aberrantly expressed miRNAs in cancer cells and the investigation of cancer-related genes controlled by miRNAs in expression. Recently, we have applied RNA-sequencing methods to generate a miRNA expression signature of PDAC and identified downregulations of 122 miRNAs in PDAC tissues [15]. Based on this signature, we have sequentially identified the tumor-suppressive miRNAs, and the genes controlled by these miRNAs, that play important roles in the molecular pathogenesis of PDAC [15].

A mature miRNA duplex is derived from the hairpin structure of pri-miRNA. The orientation of the miRNA strand determines the name of the mature miRNA form (the 5p strand derived from the 5’end of the pre-miRNA and the 3p strand arises from the 3’end). During miRNA biogenesis, one strand is preferentially loaded into Argonaute (AGO) and is considered the guide strand to function. On the other hand, the unloaded strand is called the passenger strand and degraded. However, the proportion of 5p or 3p strands incorporated into AGO is greatly dependent on the cell types or cell environments [16]. Recently, both 5p and 3p miRNA strands have been recognized as functional but have not been fully investigated in cancer cells.

The salient point of our RNA-sequencing-based signature is that expressions of some passenger strands of miRNAs derived from pre-miRNAs were significantly dysregulated in PDAC tissues. Our ectopic expression assays demonstrated that miR-216a-3p, miR-216b-3p, miR-148a-5p and miR-130b-5p acted as tumor-suppressive miRNAs in PDAC cells by targeting several oncogenes [15,17,18]. In this study, we focused on miR-30a-3p, the passenger strand derived from pre-mir-30a, based on our analysis of the signature that revealed significantly reduced levels of miR-30a-3p in PDAC tissues. Moreover, compared to miR-30a-5p, the guide strand of pre-mir-30a, miR-30a-3p, has not been fully analyzed in pancreatic cancer cells.

Our data from the present study showed that the expression of miR-30a-3p inhibited the aggressive phenotype of cancer cells, suggesting that it may act as a tumor-suppressive miRNA in PDAC cells. Our miRNA target search strategy identified 102 putative targets for miR-30a-3p regulation in PDAC cells. Importantly, expression levels of 10 of these genes (*EPS8*, *HMGA2*, *ENDOD1*, *SLC39A10*, *TGM*, *MGLL*, *SERPINE1*, *ITGA2*, *DTL,* and *UACA*) were independent prognostic factors in the multivariate analysis of survival of patients with PDAC (*p* < 0.01). We also investigated the oncogenic function of *ITGA2* in PDAC cells for this study.

## 2. Results

### 2.1. Downregulation of miR-30a-5p and miR-30a-3p in PDAC Cells and Their Tumor-Suppressive Roles in PDAC Cell Lines

Expressions of miR-30a-5p and miR-30a-3p were significantly downregulated in PDAC clinical specimens (*p* = 0.0045 and *p* = 0.0015, respectively) and in two PDAC cell lines (PANC-1 and SW1990; Figure 1A). Pearson’s analysis showed a positive correlation between miR-30a-5p and miR-30a-3p expression levels in the clinical samples (*r* = 0.7672, *p* = 0.025; Figure 1B). The clinical features of patients with PDAC considered in this study are shown in Appendix A.

Restoring expression of both miR-30a-5p and miR-30a-3p by transfection was associated with reduced malignant phenotypes of PANC-1 and SW1990 cells, as evidenced by suppressed cell proliferation, migration, and invasion compared to untransfected cells (Figure 1C,D). Meanwhile, increased expression of miR-30a-5p alone suppressed only cell migration and invasion (Figure 1C,D). Given the more substantial tumor-suppressive effect of miR-30a-3p, we focused on this miRNA in subsequent studies.

### 2.2. Identification of Genes Targeted by miR-30a-3p in PDAC Cells

To identify putative targets of miR-30a-3p regulation in PDAC cells, we assessed three datasets: (i) the TargetScan database, to identify putative targets of miR-30a-3p *in silico*; (ii) gene expression data for genes that were downregulated in miR-30a-3p-transfected PDAC cells; and (iii) gene expression data for genes that were upregulated in PDAC clinical specimens. A total of 102 genes were identified as putative targets of miR-30a-3p regulation in PDAC cells (Table 1).

### 2.3. Clinical Significance of miR-30a-3p Target Genes in PDAC

Among the 102 genes, expression levels of 12 genes (*EPS8*, *HMGA2*, *ENDOD1*, *SLC39A10*, *TGM2*, *MGLL*, *SERPINE1*, *ITGA2*, *DTL*, *UACA*, *FGD6,* and *RTP4*) had significant impacts on the prognosis of PDAC patients (Figure 2; *p* < 0.01). We validated the expression levels of these 12 genes and found that, except for *HMGA2*, all had upregulated expressions in PDAC clinical specimens (Figure 3).

Among the putative target genes for miR-30a-3p regulation in PDAC cells, high expression levels of 12 genes (*EPS8*, *HMGA2*, *ENDOD1*, *SLC39A10*, *TGM2*, *MGLL*, *SERPINE1*, *ITGA2*, *DTL*, *UAUC, FGD6* and *RTP4*) significantly predicted a worse prognosis in patients with PDAC (*p* < 0.01). Kaplan–Meier curves of the 5-year overall survival for each gene are presented.

Expression levels of 12 genes that are targets of miR-30a-3p regulation (Figure 2) were evaluated by analysis of the GSE15471 dataset. Expressions of all genes were upregulated in PDAC tissues (*n* = 39) compared to normal tissues (*n* = 39).

Moreover, expression levels of 10 genes (*EPS8*, *HMGA2*, *ENDOD1*, *SLC39A10*, *TGM*, *MGLL*, *SERPINE1*, *ITGA2*, *DTL,* and *UACA*) were identified as independent prognostic factors in multivariate analyses for the survival of patients with PDAC (*p* < 0.01; Figure 4). In advance to the multivariate analysis, the following factors were selected, and a univariable analysis was performed: age, gender, tumor stage, LN stage, metastatic stage, pathological stage, tumor size, and alcohol history. Among these factors, the tumor stage, LN stage, and pathological stage affected the prognosis of the PDAC patients. From this result, we performed the multivariate analysis with the following four factors: tumor stage, LN stage, pathological stage, and gene expression.

Multivariate analysis revealed that expression levels of 10 genes (*EPS8*, *HMGA2*, *ENDOD1*, *SLC39A10*, *TGM2*, *MGLL*, *SERPINE1*, *ITGA2*, *DTL* and *UAUC*) were independent prognostic factors for 5-year overall survival after adjustment for tumor stage, lymph node metastasis, and pathological stage (*p* < 0.01).

We investigated whether expression levels of these 10 target genes would be suppressed by the transfection of PDAC cells with miR-30a-3p. After transfection of the PDAC cell line PANC-1 with miR-30a-3p, the expression levels of all 10 of these genes were reduced (Figure 5).

Expression levels of the 10 genes (*EPS8*, *HMGA2*, *ENDOD1*, *SLC39A10*, *TGM2*, *MGLL*, *SERPINE1*, *ITGA2*, *DTL* and *UAUC*) were reduced by miR-30a-3p transfection to PANC-1 cells (48 h after the transfection). *GAPDH* was used as an internal control.

### 2.4. Direct Regulation of ITGA2 Expression by miR-30a-3p in PDAC Cells

Using dual-luciferase reporter assays involving plasmid vectors carrying partial sequences of the *ITGA2* 3’-UTR, we determined whether miR-30a-3p can directly bind to *ITGA2* in PDAC cells. We tested both the “Wild-type” *ITGA2* 3’-UTR sequence that contains the predicted miR-30a-3p target site and the “Deletion-type” sequence lacking the target site (Figure 6). A significant decrease in luciferase activity was seen in cells that were co-transfected with miR-30a-3p and the “Wild-type” vector, whereas no change in luciferase activity was observed for cells transfected with the “Deletion-type” vector and miR-30a-3p (Figure 6). These results indicate that miR-30a-3p directly binds to the 3’-UTR of *ITGA2*.

Together, these results show that the expression of *ITGA2*/ITGA2 could be suppressed by miR-30a-3p transfection (Figure 5 and Figure 6).

### 2.5. Functional Significance of ITGA2 in PDAC Cells

To confirm the oncogenic functions of *ITGA2* in PDAC cells, we performed knockdown assays using two types of siRNAs: si-*ITGA2*-1 and si-*ITGA2*-2. After confirming knockdown of *ITGA2*/ITGA2 by siRNA transfection of PANC-1 cells (Figure 7A,B), we showed that cells with siRNA-mediated knockdown of *ITGA2* gene expression suppressed cell migration and invasion activity, but not cell proliferation (Figure 7C,D). Similarly, *ITGA2* knockdown assays showed that cancer cell migration and invasive abilities were significantly suppressed in the SW1990 cells (Figure 8A–D).

We analyzed the ITGA2-related molecules with Ingenuity Pathway Analysis (IPA) software (Appendix A). Many of the molecules were associated with the migration and invasion of cells, respectively (Appendix A). These data indicate that ITGA2, in combination with the connected molecules, induces cancer cell migration and invasion firmly.

### 2.6. Overexpression of ITGA2 in Immunohistochemical Staining of PDAC Clinical Specimens

Using immunohistochemistry, we confirmed the expression of the ITGA2 and ITGB1 proteins, binding partner of ITGA2, in clinical specimens from patients with PDAC. Overexpression of both ITGA2 and ITGB1 was detected in these cancer lesions (Figure 9).

## 3. Discussion

Aberrantly expressed miRNAs in cancer cells are known to contribute to malignant transformation, metastasis, and treatment resistance [19]. In the last decade, genomics analysis methods have been actively used to identify miRNAs aberrantly expressed in various cancer cells [20,21]. Our miRNA expression profile generated by RNA sequencing provided us with new information about dysregulated miRNAs in PDAC cells [22,23,24]. Our miRNA signatures have revealed that some passenger strands of miRNAs were significantly downregulated in cancerous tissues, including miR-30a-3p and miR-30c-2-3p [25,26,27]. Recently, other groups reported that passenger strands of other miRNAs are functional in cancer cells [28,29,30,31]. Together, these findings strongly suggest that analyses that consider both strands of miRNAs are essential for cancer research.

In the human genome, mir-30a and mir-30c-2 form a cluster of 4 miRNAs on chromosome 6q13: miR-30a-5p, miR-30a-3p, miR-30c-2-5p, and miR-30c-2-3p [32,33,34]. The seed sequences of miR-30a-5p and miR-30c-2-5p are identical (GUAAACA), suggesting that the genes they control are similar. On the other hand, the seed sequences of miR-30a-3p (UUUCAGU) and miR-30c-2-3p (UGGGAGA) differ, suggesting that they control different genes. In this study, we demonstrated that the expression of miR-30a-3p significantly inhibited the malignant phenotypes of cancer cells, such as cell proliferation, migration, and invasion.

To our knowledge, no studies have reported a tumor-suppressive function of miR-30a-3p in PDAC cells. Previous studies showed the tumor-suppressive function of miR-30a-3p in several types of cancers, including gastric cancer, lung adenocarcinoma, and renal cell carcinoma [35,36,37]. Recent studies showed that some non-coding RNAs, such as Linc00483, Linc00460, and Linc01436, adsorbed miR-30a-3p and weakened the tumor-suppressive effects in gastric cancer, nasopharyngeal carcinoma, and non-small cell lung cancer, respectively [38,39,40]. In esophageal squamous cell carcinoma, expressions of both strands of miR-30a-5p and miR-30a-3p were downregulated and the low expression levels enhanced cancer cell proliferation by activating the WNT signaling pathway [41]. A recent study has shown that the WNT pathway is closely involved in lymph node metastasis-positive PDAC patients, and possibility of novel immune-based therapeutic strategies targeting WNT [42]. Our present study demonstrated that the ectopic expression of miR-30a-5p inhibited cancer cell proliferation. In contrast, the expression of miR-30a-5p did not significantly affect cell migration and invasion in PDAC cells. A previous study demonstrated that miR-30a-5p expression was downregulated in PDAC clinical specimens, and ectopic expression of miR-30a-5p increases the sensitivity of PDAC cells to gemcitabine through targeting of *FOXD1* [43]. Previous reports, as well as our present data, indicate that downregulation of both strands of miRNAs derived from pre-mir-30a are involved in malignant transformation of PDAC cells.

We also elucidated which genes associated with the promotion of PDAC are regulated by these tumor-suppressive miRNAs. In this study, we identified 135 genes (33 genes as miR-30a-5p targets and 102 genes as miR-30a-3p targets) as putative targets for pre-mir-30a regulation in PDAC cells. Among the miR-30a-3p targets, expression levels of 10 genes (*EPS8*, *HMGA2*, *ENDOD1*, *SLC39A10*, *TGM*, *MGLL*, *SERPINE1*, *ITGA2*, *DTL* and *UACA*) significantly predicted the 5-year overall survival rates of patients with PDAC (*p* < 0.01). However, prospective studies are essential to confirm the effectiveness of these genes as prognostic markers. It is also an important issue to unify and analyze the clinical setting of the patients (i.e., nodal status, TNM, grade and histological subtype, R0 or R1 if surgery was performed, neo- or adjuvant therapy received, etc.).

Notably, the expression of *EPS8* (epidermal growth factor receptor kinase substrate 8) is a highly effective prognostic marker for patients with PDAC. Our recent study of PDAC showed that the expression of *EPS8* was directly regulated to tumor-suppressive miR-130b-5p [18]. Moreover, *EPS8* was aberrantly expressed in PDAC clinical specimens, and its overexpression enhanced the aggressive phenotype of PDAC cells [18]. Another target, *HMGA2* (high-mobility group A2), is a member of the non-histone chromosomal high-mobility group protein family and acts as a transcription factor [44,45]. Multiple studies have shown that overexpression of *HMGA2* affected malignant features in multiple types of cancer [46]. Detailed analyses of these target genes will lead to a better understanding of the malignant features of PDAC.

In this study, we focused on *ITGA2* (integrin subunit alpha 2) and showed that miR-30a-3p directly regulates this gene in PDAC cells. Integrins are transmembrane receptors composed of one alpha subunit and one beta subunit, and have affinities for different extracellular membrane components [47]. Integrins activate several signal transduction pathways, and aberrant expressions of integrins have been detected in several types of cancers; overexpression of integrins can enhance cancer cell development and progression [48,49,50,51]. ITGA2 forms a homodimer with ITGB1, and dysregulated levels of ITGA2/ITGB1 can mediate the signal pathways that enhance malignant transformation in multiple types of cancer cells [52,53,54,55]. In hepatocellular carcinoma cells, ITGA2 inhibits MST1 kinase phosphorylation and activates YAP pro-oncogenic activities [52]. In PDAC cells, ITGA2 promotes PDAC cell progression through activation of a focal adhesion pathway [53]. In ovarian cancer cells, expressions of ITGA2 enhance AKT phosphorylation and further accelerate the phosphorylation of the oncogenic protein FOXO1. Moreover, a knockdown assay of *ITGA2* has shown that sensitivity of paclitaxel is improved in paclitaxel-resistant ovarian cancer cells [54]. Furthermore, ITGA2 controls the MAPK pathways and EMT in gastric cancer cells, and these events closely contribute to the chemo-resistance of gastric cancer [55]. A recent study has shown that ITGA2 increases the expression of PD-L1 by activating the STAT3 pathway in pancreatic cancer [56]. These studies indicate that overexpression of ITGA2 activates various molecular pathways and important effects on malignant transformation of cancer cells. Our recent study of PDAC revealed that the tumor-suppressive miR-124-3p directly regulates *ITGA3* and *ITGB1* expression, and aberrant expressions of these integrins were closely involved in PDAC molecular pathogenesis [57]. Taken together, tumor-suppressive miRNAs controls the expression of ITGA2/ITGB1 and ITGA3/ITGB1 in PDAC cells, and the aberrant expressions of these integrins can play pivotal roles in the malignant features of PDAC. Thus, signaling pathways that are activated by ITGA2/ITGB1 and ITGA3/ITGB1 could be therapeutic targets for PDAC.

## 4. Materials and Methods

### 4.1. Collection of Clinical Human PDAC Specimens, Pancreas Tissue Specimens, and PDAC Cell Lines

The present study was approved by the Bioethics Committee of Kagoshima University (Kagoshima, Japan; approval no. 160,038 28-65). Written prior informed consent and approval were obtained from all of the patients.

In this study, 31 PDAC clinical samples were collected from patients with PDAC who underwent resection at Kagoshima University Hospital from 1997 to 2016. Fifteen normal pancreatic tissue specimens were collected from noncancerous regions. The clinical samples were staged according to the American Joint Committee on Cancer/Union Internationale Contre le Cancer (UICC) TNM classification. Clinical features in PDAC specimens are shown in Appendix A.

PDAC cell lines SW1990 and PANC-1 were purchased from the American Type Culture Collection (Manassas, VA, USA) and RIKEN Cell Bank (Tsukuba, Ibaraki, Japan), respectively.

### 4.2. RNA Extraction and Quantitative Real-Time Reverse Transcription Polymerase Chain Reaction (qRT-PCR)

The methods for RNA extraction from clinical specimens and cell lines, and qRT-PCR have been described previously [25,26,55]. TaqMan probes and primers used in this study are listed in Appendix A.

### 4.3. Transfection of PDAC Cells with miRNAs, siRNAs, and Plasmid Vectors

The procedures for transfection of PDAC cells with miRNAs, siRNAs, and plasmid vectors were described previously [25,26,55]. The reagents used in this study are listed in Appendix A.

### 4.4. Functional Assays in PDAC Cells (Cell Proliferation, Migration, and Invasion Assays)

The procedures for functional assays in cancer cells (proliferation, migration, and invasion) are described in our previous studies [25,26,55]. Cells were transfected with 10 nM miRNAs or siRNAs.

Cell proliferation was evaluated with XTT assays. Migration assays were performed with uncoated trans-well polycarbonate membrane filters, and invasion assays were carried out using modified Boyden chambers.

### 4.5. Identification of the miR-30a-5p and miR-30a-3p Targets in PDAC Cells

We selected putative target genes having binding sites for miR-30a-5p and miR-30a-3p using TargetScanHuman ver.7.2 (http://www.targetscan.org/vert_72/; data was downloaded on 13 July 2018). Our microarray data (miR-30a-5p or miR-30a-3p transfected cells) were deposited in the GEO repository under accession number GSE113066. To examine upregulated genes in PDAC clinical specimens, expression data was obtained from the GEO database (GSE15471).

### 4.6. Clinical Database Analysis of miRNA Target Genes in PDAC Clinical Specimens

For analysis of differential gene expression between normal tissues and cancer tissues, we utilized GSE15471 datasets obtained from the Gene Expression Omnibus (GEO). Briefly, GSE15471 contains mRNA array data from 36 PDAC tumors and matching normal pancreatic tissue samples. The data was obtained using Affymetrix U133 Plus 2.0 whole-genome chips. Expression levels are shown in signal intensities, and for genes that had multiple probes, the mean value was used.

For the Kaplan–Meier survival analysis, we downloaded TCGA clinical data (TCGA, Firehose Legacy) from cBioportal (https://www.cbioportal.org). Gene expression grouping data for each gene was collected from OncoLnc (http://www.oncolnc.org). R version 4.0.2 (R Foundation for Statistical Computing, Vienna, Austria) was used for statistical analyses.

### 4.7. Plasmid Construction and Dual-Luciferase Reporter Assays

Plasmid vectors, including vectors carrying *ITGA2* with the wild-type sequences of the miR-30a-3p binding sites in the 3′-UTR or with these sequences deleted, were prepared. The procedures for transfection and dual-luciferase reporter assays were described in our previous studies [25,26,55]. The reagents used in this study are listed in Appendix A.

### 4.8. Western Blotting and Immunohistochemistry

The procedures for western blotting and immunohistochemistry were described in our previous studies [25,26,55]. The antibodies used in this study are listed in Appendix A.

### 4.9. Downregulation of miR-30a-5p and miR-30a-3p in PDAC Cells and Their Tumor-Suppressive Roles in PDAC Cell Lines

Mann–Whitney U tests were applied for comparisons between two groups. To compare multiple groups, one-way analysis of variance and Dunnett’s test were applied. These analyses were carried out using JMP Pro 14 (SAS Institute Inc., Cary, NC, USA).

## 5. Conclusions

Analysis of our PDAC miRNA expression signature revealed that expression of both strands of pre-mir-30a (miR-30a-5p and miR-30a-3p) was downregulated in PDAC tissues. Ectopic expression assays showed that both miRNAs act as tumor-suppressive miRNAs in PDAC cells. A total of 102 genes were identified as targets for control by miR-30a-3p in PDAC cells. Among these targets, the expression levels of 10 genes (*EPS8*, *HMGA2*, *ENDOD1*, *SLC39A10*, *TGM*, *MGLL*, *SERPINE1*, *ITGA2*, *DTL,* and *UACA*) significantly predicted the 5-year overall survival rates of patients with PDAC (*p* < 0.01). Aberrant expression of *ITGA2* contributed to the malignant transformation of PDAC cells. Our miRNA-based analysis strategy provides important insights into the role of miRNA in the molecular pathogenesis of PDAC.

## Figures and Tables

**Figure 1 ijms-21-06459-f001:**
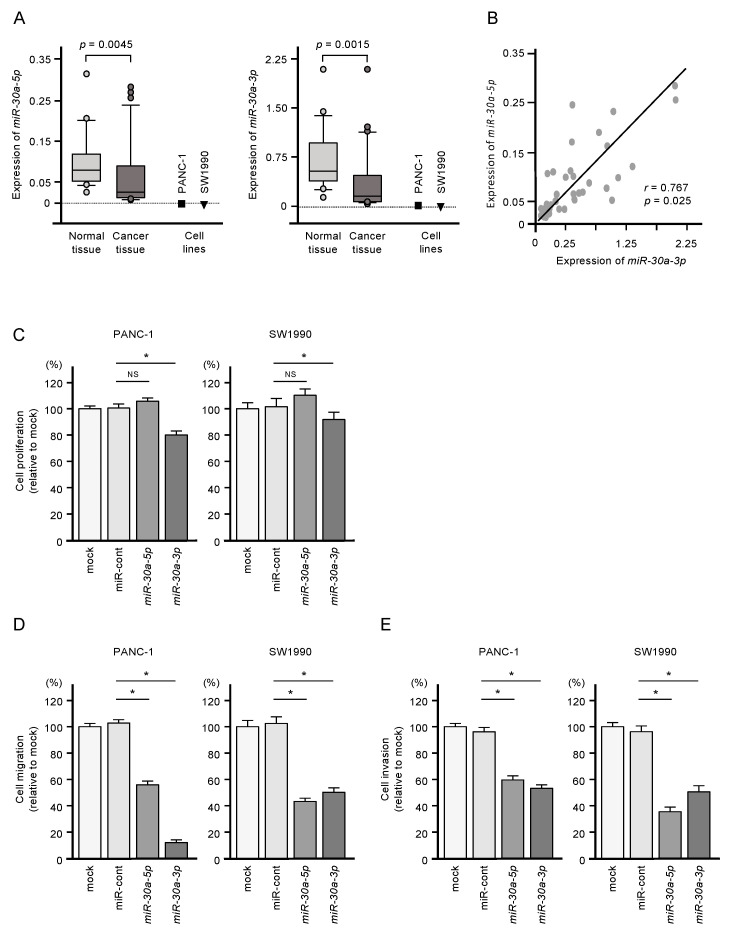
Tumor-suppressive functions of miR-30a-5p and miR-30a-3p in pancreatic ductal adenocarcinoma (PDAC) cells. (**A**) Expression levels of miR-30a-5p and miR-30a-3p in PDAC clinical specimens and cell lines (PANC-1 and SW1990). Data were normalized relative to the expression of *RNU48.* (**B**) Pearson’s coefficient showed positive correlations between the expression levels of miR-30a-5p and miR-30a-3p in clinical specimens. (**C**) Cell proliferation assessed using XTT assays. Data were collected 72 h after miRNA transfection (* *p* < 0.0001). (**D**) Cell migration assessed with a membrane culture system. Data were collected 48 h after seeding the cells into the chambers (* *p* < 0.0001). (**E**) Cell invasion determined using Matrigel invasion assays conducted 48 h after the seeding of the miRNA-transfected cells into the chambers (* *p* < 0.0001).

**Figure 2 ijms-21-06459-f002:**
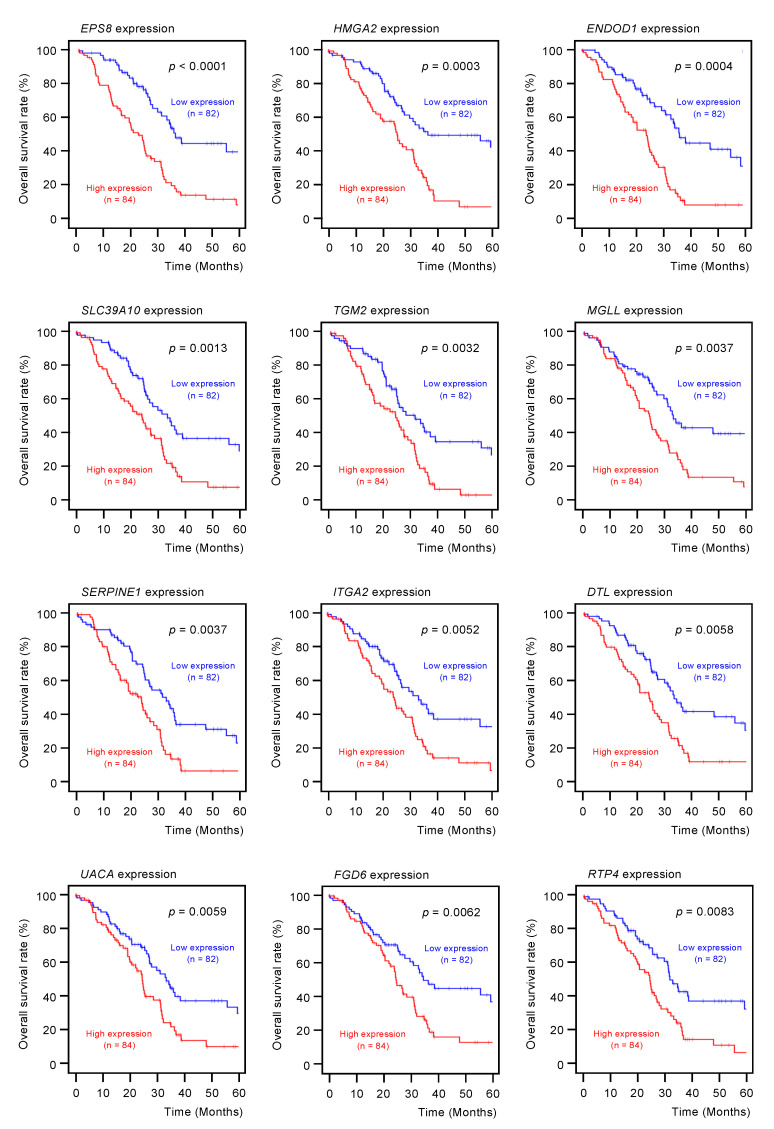
Clinical significance of the miR-30a-3p target genes by TCGA database analysis.

**Figure 3 ijms-21-06459-f003:**
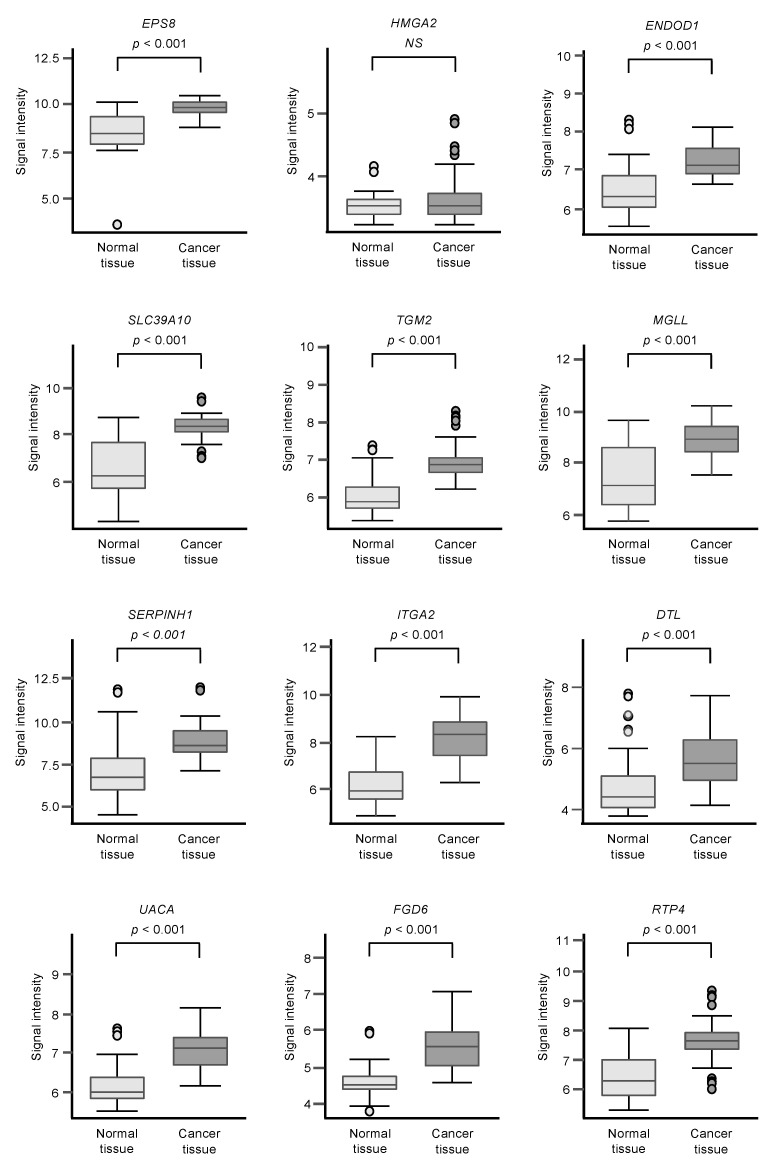
GSE15471 dataset analyses of the expression levels of 12 genes (predicted 5-year survival) that are targets for miR-30a-3p regulation in PDAC clinical specimens.

**Figure 4 ijms-21-06459-f004:**
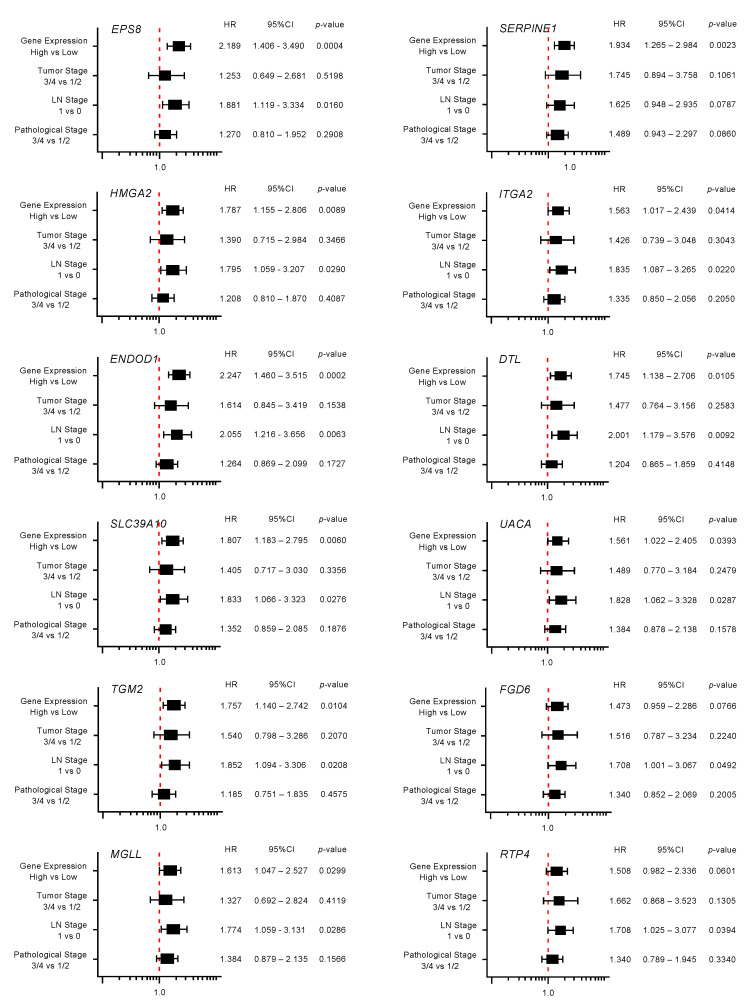
Forest plot of the multivariate analysis of the 12 genes that are targets of miR-30a-3p regulation (predicted 5-year survival).

**Figure 5 ijms-21-06459-f005:**
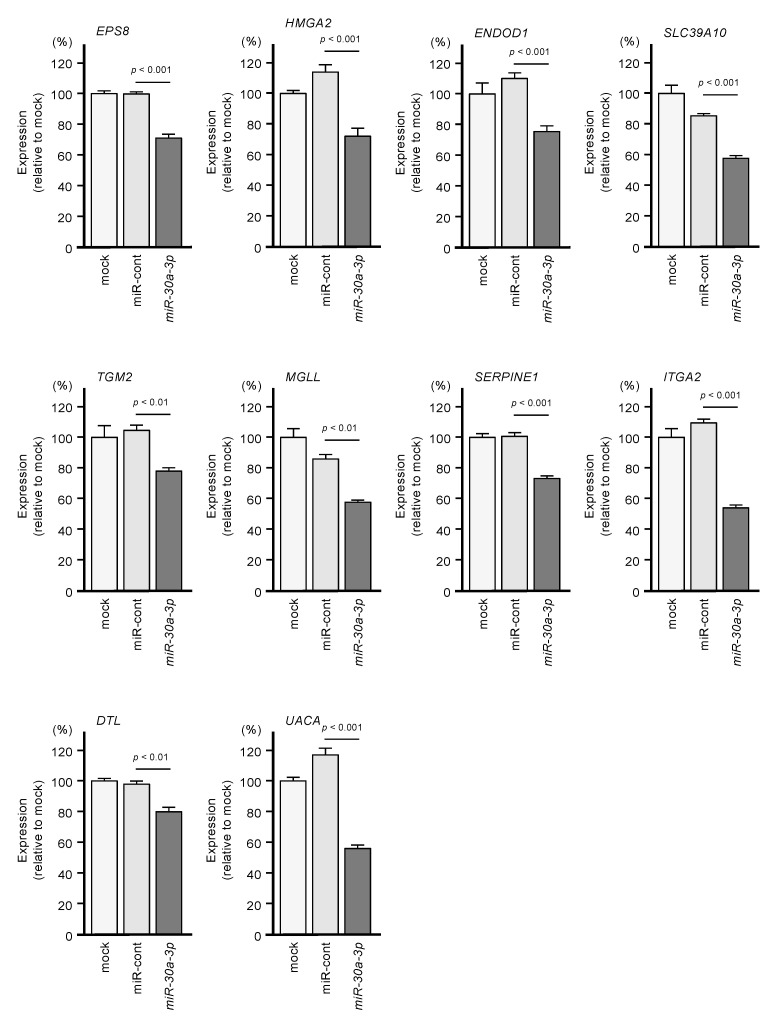
Regulation of the expression of the 10 target genes following transfection of PANC-1 cells with miR-30a-3p.

**Figure 6 ijms-21-06459-f006:**
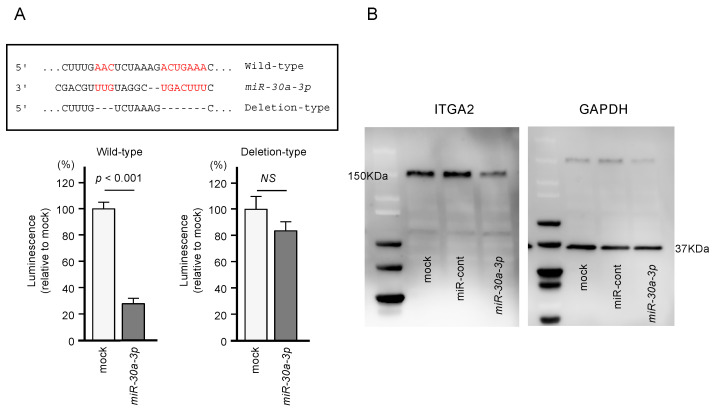
Direct regulation of *ITGA2* by miR-30a-3p in PANC-1 cells. (**A**) The TargetScan database showed that one putative binding site of miR-30a-3p was annotated in the 3′-UTR of *ITGA2*. Dual luciferase reporter assays showed that luminescence activities were reduced by co-transfection of PANC-1 cells with wild-type vector (containing miR-30a-3p binding sites) and miR-30a-3p. Normalized data were calculated as *Renilla*/firefly luciferase activity ratios. (**B**) Protein expression levels of ITGA2 were significantly reduced by miR-30a-3p transfection to PANC-1 cells (48 h after the transfection). Whole Western blotting images are shown. GAPDH was used as an internal control.

**Figure 7 ijms-21-06459-f007:**
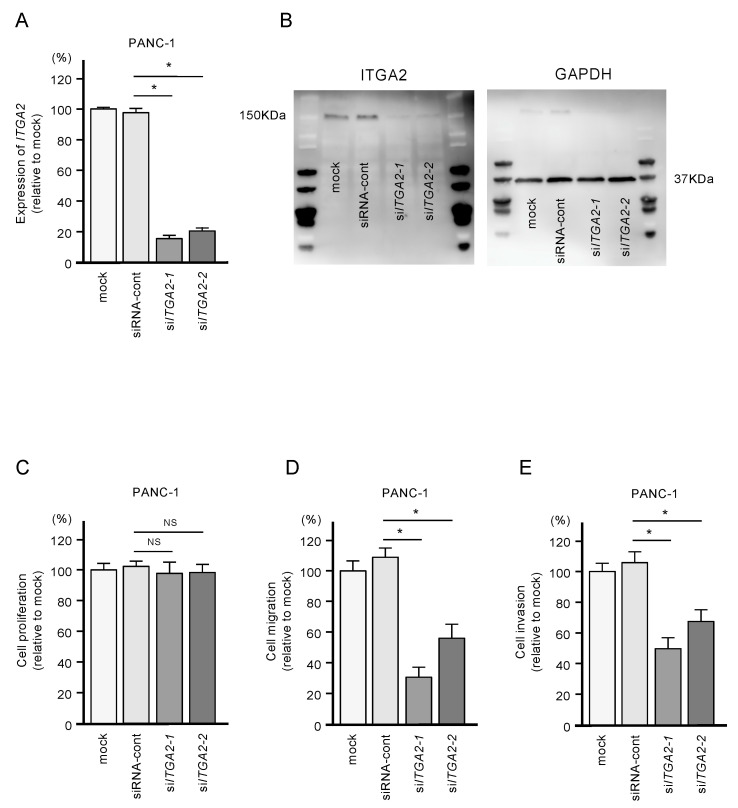
Effects of *ITGA2* knockdown in PANC-1 cells. (**A**) mRNA and (**B**) protein expression levels of *ITGA2* after transfection of two types of si*ITGA2* (si*ITGA2-1* and si*ITGA2-2*; 72 h after transfection). *GAPDH*/GAPDH was used as an internal control. (**C**) Cell proliferation using XTT assays. Data were collected at 72 h after miRNA transfection. (**D**) Cell migration with a membrane culture system. Data were collected at 48 h after seeding the cells into the chambers (* *p* < 0.0001). (**E**) Cell invasion using Matrigel invasion assays at 48 h after seeding miRNA-transfected cells into the chambers. (* *p* < 0.0001)

**Figure 8 ijms-21-06459-f008:**
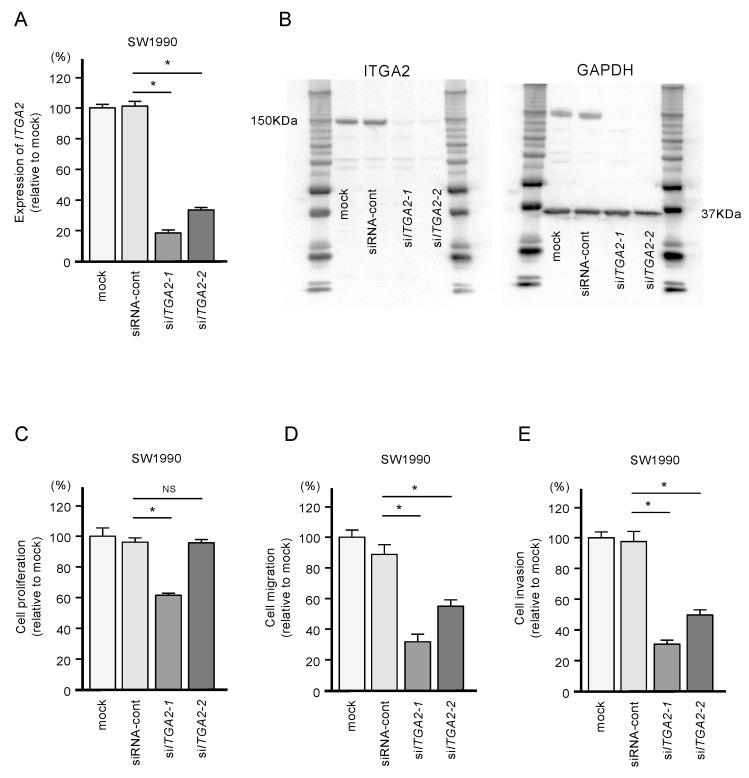
Effects of *ITGA2* knockdown in SW1990 cells. (**A**) mRNA and (**B**) protein expression levels of *ITGA2* by transfection with two types of si*ITGA2* in SW1990 cells. (**C,D,E**) Cell proliferation, migration and invasion assays in SW1990 cells. (* *p* < 0.0001)

**Figure 9 ijms-21-06459-f009:**
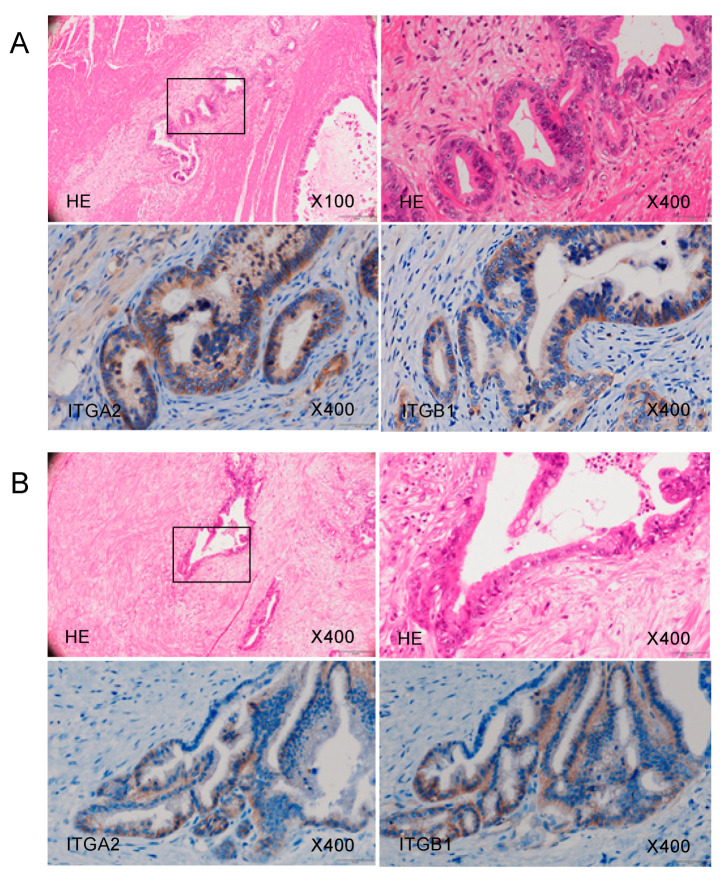
ITGA2 overexpression in clinical PDAC specimens. Representative immunohistochemical images with staining of ITGA2 and ITGB1 in clinical samples. Overexpression of ITGA2 and ITGB1 was detected in the cancer lesions. Images of samples from Patient (**A**) number 20 and (**B**) number 17 (Appendix A) are shown.

**Table 1 ijms-21-06459-t001:** Identification of putative targets regulated by miR-30a-3p in PDAC cells.

Entrez Gene ID	Gene Symbol	Gene Name	Expression in PANC-1 miR-30a-3p Transfectants (FClog_2_ < −1.0)	GSE15471 (FClog2 > 1.0)	TCGA OncoLnc OS *p*-Value (In 5 Years)
2059	*EPS8*	epidermal growth factor receptor pathway substrate 8	−1.4210253	1.262866923	0.0001
8091	*HMGA2*	high mobility group AT-hook 2	−1.3260773	1.319899625	0.0003
23052	*ENDOD1*	endonuclease domain containing 1	−1.023117	1.177174988	0.0004
57181	*SLC39A10*	solute carrier family 39 (zinc transporter), member 10	−1.305532	1.770031909	0.0013
7052	*TGM2*	transglutaminase 2	−1.1769093	2.232351312	0.0032
11343	*MGLL*	monoglyceride lipase	−2.0041718	1.656630451	0.0037
5054	*SERPINE1*	serpin peptidase inhibitor, clade E (nexin, plasminogen ativator inhibitor type 1), member 1	−1.8854839	1.966089688	0.0037
3673	*ITGA2*	integrin, alpha 2 (CD49B, alpha 2 subunit of VLA-2 receptor)	−2.235423	2.619190426	0.0052
51514	*DTL*	denticleless E3 ubiquitin protein ligase homolog (Drosophila)	−1.4902039	1.119465213	0.0058
55075	*UACA*	uveal autoantigen with coiled−coil domains and ankyrin repeats	−2.600379	1.2785210	0.0059
55785	*FGD6*	FYVE, RhoGEF and PH domain containing 6	−1.1297837	1.472426111	0.0062
64108	*RTP4*	receptor (chemosensory) transporter protein 4	−1.6157867	1.179594488	0.0083
118429	*ANTXR2*	anthrax toxin receptor 2	−1.5205879	1.927129062	0.0101
55704	*CCDC88A*	coiled-coil domain containing 88A	−1.9858294	1.104710907	0.0112
5358	*PLS3*	plastin 3	−1.2563933	1.111472402	0.0144
25907	*TMEM158*	transmembrane protein 158 (gene/pseudogene)	−1.7876587	2.088651268	0.0178
79026	*AHNAK*	AHNAK nucleoprotein	−1.2770338	1.085859303	0.0179
54739	*XAF1*	XIAP associated factor 1	−1.237152	1.868846867	0.018
26509	*MYOF*	myoferlin	−1.5663022	2.424595363	0.019
5782	*PTPN12*	protein tyrosine phosphatase, non-receptor type 12	−1.2349758	1.501494024	0.0205
55450	*CAMK2N1*	calcium/calmodulin-dependent protein kinase II inhibitor 1	-1.5350429	1.530841589	0.0265
22822	*PHLDA1*	pleckstrin homology-like domain, family A, member 1	−2.3576307	1.629925658	0.031
9976	*CLEC2B*	C-type lectin domain family 2, member B	−1.2099471	1.787926875	0.0393
3217	*HOXB7*	homeobox B7	−1.9851137	1.546718079	0.0423
301	*ANXA1*	annexin A1	−1.7945881	2.109602404	0.0441
285761	*DCBLD1*	discoidin, CUB and LCCL domain containing 1	−1.3696324	1.736051328	0.0452
3656	*IRAK2*	interleukin-1 receptor-associated kinase 2	−2.023258	1.139261425	0.0527
7074	*TIAM1*	T−cell lymphoma invasion and metastasis 1	−1.0873771	1.191799754	0.0567
91404	*SESTD1*	SEC14 and spectrin domains 1	−1.0042286	1.389203339	0.0615
9770	*RASSF2*	Ras association (RalGDS/AF-6) domain family member 2	−1.1753412	1.512689587	0.0623
2113	*ETS1*	v-ets avian erythroblastosis virus E26 oncogene homolog 1	−1.5893964	1.15605574	0.0733
6566	*SLC16A1*	solute carrier family 16 (monocarboxylate transporter), member 1	−1.7025123	1.289199126	0.0748
7421	*VDR*	vitamin D (1,25- dihydroxyvitamin D3) receptor	−1.0838604	1.641355784	0.0829
54810	*GIPC2*	GIPC PDZ domain containing family, member 2	−1.1281186	1.357104573	0.0863
56937	*PMEPA1*	prostate transmembrane protein, androgen induced 1	−1.1079406	2.165168297	0.0901
55686	*MREG*	melanoregulin	−2.0134192	1.110134893	0.0916
7046	*TGFBR1*	transforming growth factor, beta receptor 1	−1.5660658	1.567228059	0.101
57713	*SFMBT2*	Scm-like with four mbt domains 2	−2.1900904	1.273632017	0.107
4642	*MYO1D*	myosin ID	−1.0740082	1.117494381	0.113
4131	*MAP1B*	microtubule-associated protein 1B	−1.5398299	1.867512419	0.122
3397	*ID1*	inhibitor of DNA binding 1, dominant negative helix-loop-helix protein	−1.0427207	1.289751677	0.134
25927	*CNRIP1*	cannabinoid receptor interacting protein 1	−1.5424297	1.231971168	0.147
54855	*FAM46C*	family with sequence similarity 46, member C	−1.192339	1.326045782	0.147
90459	*ERI1*	exoribonuclease 1	−1.0246572	1.087664329	0.149
7035	*TFPI*	tissue factor pathway inhibitor (lipoprotein-associated coagulation inhibitor)	−2.0193892	1.701846836	0.151
3433	*IFIT2*	interferon-induced protein with tetratricopeptide repeats 2	−1.7064413	1.460924272	0.169
10100	*TSPAN2*	tetraspanin 2	−1.0831516	1.41156579	0.169
18	*ABAT*	4-aminobutyrate aminotransferase	−1.3639479	1.797952852	0.174
27286	*SRPX2*	sushi-repeat containing protein, X−linked 2	−1.1989174	2.472950218	0.182
9456	*HOMER1*	homer homolog 1 (Drosophila)	−1.0073853	1.299511365	0.191
3696	*ITGB8*	integrin, beta 8	−1.4881207	1.573789775	0.203
2633	*GBP1*	guanylate binding protein 1, interferon-inducible	−1.7881731	1.911867379	0.215
83700	*JAM3*	junctional adhesion molecule 3	−1.1177526	1.065465122	0.217
7498	*XDH*	xanthine dehydrogenase	−2.073526	1.186465821	0.22
10846	*PDE10A*	phosphodiesterase 10A	−1.1002035	1.167788231	0.223
11031	*RAB31*	RAB31, member RAS oncogene family	−1.2654961	2.855210292	0.225
135228	*CD109*	CD109 molecule	−2.4484148	2.363606904	0.231
64105	*CENPK*	centromere protein K	−1.186185	1.179594934	0.233
57045	*TWSG1*	twisted gastrulation BMP signaling modulator 1	−2.5573916	1.19989424	0.234
22891	*ZNF365*	zinc finger protein 365	−1.1564493	1.11269613	0.256
2635	*GBP3*	guanylate binding protein 3	−1.5946016	1.672916793	0.26
56913	*C1GALT1*	core 1 synthase, glycoprotein-N-acetylgalactosamine 3-beta-galactosyltransferase, 1	−1.5604228	1.322049565	0.262
2697	*GJA1*	gap junction protein, alpha 1, 43kDa	−2.5297747	1.75081194	0.264
2004	*ELK3*	ELK3, ETS-domain protein (SRF accessory protein 2)	−1.2669431	1.489197891	0.264
5125	*PCSK5*	proprotein convertase subtilisin/kexin type 5	−1.0345638	1.411697675	0.286
57157	*PHTF2*	putative homeodomain transcription factor 2	−1.0247078	1.303249973	0.287
51762	*RAB8B*	RAB8B, member RAS oncogene family	−1.4737062	1.194672398	0.299
3176	*HNMT*	histamine N-methyltransferase	−1.017499	1.193174738	0.327
9644	*SH3PXD2A*	SH3 and PX domains 2A	−1.789658	1.631376261	0.362
143903	*LAYN*	layilin	−1.4523163	1.461949991	0.368
8876	*VNN1*	vanin 1	−2.3857656	2.077747375	0.395
11329	*STK38*	serine/threonine kinase 38	−1.2525054	1.275932579	0.403
9890	*LPPR4*	Lipid phosphate phosphatase-related protein type 4	−2.102301	1.232443047	0.412
2313	*FLI1*	Fli-1 proto-oncogene, ETS transcription factor	−1.902698	1.132852853	0.428
6925	*TCF4*	transcription factor 4	−1.1343423	1.6263018	0.43
10487	*CAP1*	CAP, adenylate cyclase-associated protein 1 (yeast)	−1.3670303	1.280092932	0.456
10687	*PNMA2*	paraneoplastic Ma antigen 2	−1.3189737	2.060441736	0.471
55790	*CSGALNACT1*	chondroitin sulfate N-acetylgalactosaminyltransferase 1	−1.5380203	1.083904879	0.482
79772	*MCTP1*	multiple C2 domains, transmembrane 1	−2.0109892	1.206431617	0.483
89795	*NAV3*	neuron navigator 3	−1.9344572	1.513845893	0.485
135892	*TRIM50*	tripartite motif containing 50	−1.2579012	1.230273363	0.494
83716	*CRISPLD2*	cysteine-rich secretory protein LCCL domain containing 2	−1.025413	1.461831106	0.494
6558	*SLC12A2*	solute carrier family 12 (sodium/potassium/chloride transporter), member 2	−2.2465677	1.117656881	0.524
26064	*RAI14*	retinoic acid induced 14	−1.1299313	1.438363973	0.562
6443	*SGCB*	sarcoglycan, beta (43 kDa dystrophin-associated glycoprotein)	−2.1484566	1.119623109	0.624
92241	*RCSD1*	RCSD domain containing 1	−1.6868691	1.19300128	0.645
3106	*HLA-B*	major histocompatibility complex, class I, B	−1.2878256	1.410905751	0.658
5592	*PRKG1*	protein kinase, cGMP-dependent, type I	−1.2409348	1.102178721	0.668
9590	*AKAP12*	A kinase (PRKA) anchor protein 12	−1.5021276	1.672466982	0.689
493869	*GPX8*	glutathione peroxidase 8 (putative)	−1.1502694	2.136870238	0.699
285203	*EOGT*	EGF domain-specific O-linked N-acetylglucosamine (GlcNAc) transferase	−1.5562044	1.219417735	0.711
11010	*GLIPR1*	GLI pathogenesis-related 1	−1.0720639	1.936762677	0.733
5167	*ENPP1*	ectonucleotide pyrophosphatase/phosphodiesterase 1	−1.0345893	1.107172826	0.756
9615	*GDA*	guanine deaminase	−1.0771136	1.280713291	0.803
1687	*DFNA5*	deafness, autosomal dominant 5	−1.2998511	1.348055613	0.816
5552	*SRGN*	serglycin	−1.7418242	1.491958762	0.867
9444	*QKI*	QKI, KH domain containing, RNA binding	−1.3413125	1.553606699	0.871
84056	*KATNAL1*	katanin p60 subunit A-like 1	−1.6962346	1.164463984	0.877
151887	*CCDC80*	coiled-coil domain containing 80	−1.1713166	2.301304933	0.901
133418	*EMB*	embigin	−1.0146027	1.447293274	0.963
2124	*EVI2B*	ecotropic viral integration site 2B	−1.6717868	1.54469176	0.97
79625	*NDNF*	neuron-derived neurotrophic factor	−1.3546426	1.114794439	0.983

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
