# Peer review of "Role of miR-30a-3p Regulation of Oncogenic Targets in Pancreatic Ductal Adenocarcinoma Pathogenesis"

_ijms, 2020, doi:10.3390/ijms21186459_

Round 1
Reviewer 1 Report
Comments to the author:
The current study by Shimomura et al. is an investigation of dysregulated micro RNAs in pancreatic ductal adenocarcinoma (PDAC) tissue and their potential biological functions and clinical impact on PDAC patient survival. From which, the author reported the underlying tumour suppression effect of miR-30a-3p, a significantly downregulated miRNA in PDAC, which attenuated the tumorigenic aggressiveness of PDAC cells. The author further proposed a panel of putative targeted genes of this micro RNA and did Kaplan-Meier survival analysis from an open-access data base for those genes in PDAC, and revealed 10 of them was correlated with poor survival of PDAC patient.
For the functional study, the candidate prognostic gene ITGA2 was selected, which was showed to be overexpressed in PDAC tissue. The author suggested that expression of ITGA2 was directed inhibited by miR-30a-3p, and the migration and invasion activity of cancer cells would be supressed by knocking down of ITGA2.
The study outline was well sounded, and the methodologies were applied properly. Here are some comments and questions to this study:
Major comments:
- The multivariable analysis should preclude those factors that was presented without significance at univariable analysis (DOI: 10.1016/j.ebiom.2019.04.008). The reviewer is curious about how was the Figure 4 generated? Since there is no detailed description about other factors at univariable analysis, were all of the parameters Tumor Stage, LN Stage, and Pathological Stage presented significant (p < 0.05) for every of those 10 genes? And what about other clinicopathological factors, are they significant or not, as age, tumor location? To the knowledge of the reviewer the TCGA database only provides limited information for this kind of analysis, but it is important to show those results, which can help readers have an unbiased understanding of the conclusions made by the author.
- Although the discussion part was quite broad and comprehensive, the reviewer strongly recommends a more concise discussion, focusing on the literatures related to main conclusions. For instance, Line 84 ‘More recently, our group generated miRNA expression signatures by RNA-sequencing of breast cancer and esophageal squamous cell carcinomas’, of course this information is good for showing others the research background of your group. However, this is apparently not correlated to current study, and even worse, it can bring the thoughts of unnecessary self-citation to the reviewer and potential readers.
- The reviewer understands the author were attempted to elucidate the underlying mechanisms of how ITGA2 reduced the tumorigenic phenotypes of PDAC. However, none of the 7 Figures from the results demonstrated any potential mechanism, even at the discussion part, there are just few sentences mentioned this point. i.e. Line 105 till Line 107 ‘In esophageal squamous……Wnt signaling pathway’, Line 136 ‘……STATS pathway……’. As the claim made by the author like the last sentence at discussion part ‘Thus, signaling pathways that are activated by ITGA2/ITGB1 and ITGA3/ITGB1 could be therapeutic targets for PDAC’, which is not sufficiently supported by such limited information. Thus, the reviewer recommends a comprehensive bioinformatic analysis of ITGA2/ITGB1 and ITGA3/ITGB1, to interpret the essential pathways that involved in those targets and mechanisms that can theoretically explain the phenotypes been observed in this study. This bioinformatic analysis could be performed by the Ingenuity Pathway Analysis software (IPA, https://doi.org/10.1016/j.trsl.2019.06.003 as an example), and the results could be added as figure 8. When this interactive network been built, much more clear information would stand out. Then the author would be able to discuss more interesting findings from previous related studies, and to draw logical novel hypothesis, which will make this study be a completed and solid report.
Minor comments:
The English language editing of this report could and should be improved, many obvious grammar mistakes were appeared, i.e. line 94, ‘suggesting that the genes they control are similar’ should be ‘suggesting that the control function of those genes are similar. Or suggesting that those genes have similar control function’. Line 95, ‘the seed sequences of …differ’, should be ‘the seed sequences…are different’. Therefore, the reviewer recommends that the author to seek professional English language editing service from this journal, or as help from researchers whose mother tongue is English.
Reviewer 2 Report
To the authors
Hirotaka Shimomura et al. corroborated their recent studies highlighting the fact that some passenger strands of miRNAs in the cancer molecular pathogenesis prompts the investigation of a microRNA (miRNA) expression signature. They dissected oncogenes closely involved in PDAC molecular pathogenesis, focusing on miR-30a-3p regulation. Next, they described 102 putative targets of miR-30a-3p regulation in PDAC cells, while uncovering 10 genes to be independent prognostic factors in multivariate analysis of survival of PDAC patients. The manuscript covers an interesting topic, nonetheless, there are few sections that deserve to be restructured, in order to achieve the level and comprehensive overview that a journal like IJMS would aim to.
Major points to consider in subsequent versions:
- Methods: experiments need to be repeated on 1-2 more cell lines. Alternatively, the author might also explain which in silico or publicly available approach might circumvent this point (i.e. to their knowledge do the author expect different human PDAC cell line model behavior while applying the employed in vitro approach on PANC1 (see also point 5)?
- Results, Figure 1 B: Sample size affects the power or ability of all statistical tests to detect a relationship between two variables when it truly exists. Correlation coefficient is no different and can give a false negative result if a sample size is inadequate. Hence, the smaller the correlation coefficient between two variables the investigators would like to detect in a study, the larger the sample size is required. Because Spearman’s correlation coefficient is not as efficient as Pearson’s coefficient, an extra 10% in sample size is needed to achieve the same statistical power if Spearman’s correlation coefficient is the targeted outcome instead of Pearson’s correlation coefficient. Can the author comment on this topic?
- Prognostic models. The authors analyzed differentially clinical outcome impacting gene signatures. This is fine, as long as the authors point out how the stratified the patients expression level of the considered variables (over the median vs. below the median and/or quartile). Nonetheless, as a prognostic biomarker in terms of both overall and event-free survival, the clinical characteristics of those patients can deeply impact the HR (i.e. nodal status, TNM, grade and histological subtype, R0 or R1 if surgery was performed, neo- or adjuvant therapy received, etc.). We acknowledge that this can be beyond the scope of the manuscript and anyway not possible with a retrospective in silico interrogation. Nonetheless, if co-variates suitable for multivariate statistical analyses are only partially available this should be mentioned as a study limitation, especially if hazards proportionality is not respected and COX multivariate model cannot be performed. Indeed, those are important information and should be provided in order to propose the elaborated score as potentially valuable in a clinical setting. Otherwise, this analysis can only be hypothesis-generating and the association with survival should be tuned down while highlighting the lack of validation to date.
- Moreover, the multivariate analysis can show the impact of several variables. Nonetheless, the authors examined a cohort retrospectively (or Did I miss something?).
Some key statistics cannot be measured, and significant biases may affect the selection of controls. Researchers cannot control exposure or outcome assessment and instead must rely on others for accurate record keeping. Assuming these study limitations can be worth in order to highlight the value and the limitations of the data presented, interpreting them as hypothesis-generating data that need to be confirmed in a prospective fashion.
- In Figure 4 the authors briefly allude to lymph node metastases, while adjusting the survival impact of the analyzed variables.This section can be slightly expanded, comprising novel findings of this topic and the insights about tumour immune-microenvironment, leading to the emergence of aberrant signaling pathways as critical factors modulating central gene-expression signatures that fuel pancreatic tumour both directly and indirectly, by shaping the immune milieu and driving nodal invasion (PMID: 31277479). Indeed, since down-regulation of miR-30a-3p/5p promotes cancer cell proliferation by activating the Wnt signaling pathway through inhibition of Wnt2 and Fzd2, it is tempting to speculate about the correlation between miR-30a-3p and the PDAC microenvironment, in both ex vivo, in silico and in vitro PDAC model (i.e. PANC1 cells - PMID: 31277479, see also point 1). In the frame of this thinking, it is remarkable how the MAPK/ERK pathway and epithelial-mesenchymal transition (EMT) can be also implicated upon ITGA2 knockdown, as in other solid and haematological malignancies (PMID: 32232000; PMID: 32043788), in which the adhesion system has been related to cancer invasiveness and BRAF/mapk inhibition alters the microRNA cargo.
- General comment:
While discussing (I quote) “Another target, HMGA2 (high-mobility group A2), is a member of the non-histone chromosomal high-mobility group protein family and acts as a transcription factor). The author did allude to some HMGA2 functions. Nonetheless, HMGA2, via mTOR, deeply impact on tumour angiogenesis in several malignancies (PMID: 29755672; PMID: 30755600). Therefore, an additional tempting implication of the author findings seems to point towards a potential theragnostic landscape targeting HMGA2 and mTOR inhibition at the same time, also suppressing the phosphorylation of AKT and mTOR without altering total AKT and mTOR level.
Finally, the authors' statement " ITGA2 forms a homodimer with ITGB1, and dysregulated levels of ITGA2/ITGB1 can mediate signal pathways that enhance malignant transformation in multiple types of cancer cells [50-53]" I personally miss some important translational aspect potentially related to this aspect, pointing towards a potential Achilles’ heel of PDAC that might be exploited therapeutically in the future. Indeed, tumour-stroma interactions are of key importance for PDAC progression. Cancer-associated fibroblasts (CAFs) and mast cells (MC) affected the sensitivity of PDAC cells to gemcitabine/nabpaclitaxel. The MCs have been uncovered to drive resistance to drugs by reducing apoptosis, by activating the TGF-β signaling, and mediating chemo-resistance (PMID: 30866547).
Minor
The figures would require some beautification and should be as simplified as possible in order to make them as much readable as possible.
Some linguistic improvement might be achieved by performing a native speaker revision.
Round 2
Reviewer 2 Report
I am satisfied by the authors’ response. No further comments from this reviewer.